# Rethinking Light Decoder-based Solvers for Vehicle Routing Problems

**Ziwei Huang**[1*], **Jianan Zhou**[2*], **Zhiguang Cao**[1†], **Yixin Xu**[1]
[1]School of Computing and Information Systems, Singapore Management University, Singapore
[2]College of Computing and Data Science, Nanyang Technological University, Singapore
`ziweihuang@smu.edu.sg, jianan004@e.ntu.edu.sg,`
`zgcao@smu.edu.sg, yixinxu@smu.edu.sg`

## Abstract

Light decoder-based solvers have gained popularity for solving vehicle routing problems (VRPs) due to their efficiency and ease of integration with reinforcement learning algorithms. However, they often struggle with generalization to larger problem instances or different VRP variants. This paper revisits light decoder-based approaches, analyzing the implications of their reliance on static embeddings and the inherent challenges that arise. Specifically, we demonstrate that in the light decoder paradigm, the encoder is implicitly tasked with capturing information for all potential decision scenarios during solution construction within a single set of embeddings, resulting in high information density. Furthermore, our empirical analysis reveals that the overly simplistic decoder struggles to effectively utilize this dense information, particularly as task complexity increases, which limits generalization to out-of-distribution (OOD) settings. Building on these insights, we show that enhancing the decoder capacity, with a simple addition of identity mapping and a feed-forward layer, can considerably alleviate the generalization issue. Experimentally, our method significantly enhances the OOD generalization of light decoder-based approaches on large-scale instances and complex VRP variants, narrowing the gap with the heavy decoder paradigm. Our code is available at: `https://github.com/ziweileonhuang/reld-nco`.

## 1 Introduction

Vehicle Routing Problems (VRPs) are a fundamental class of NP-hard combinatorial optimization problems (COPs) with wide-ranging applications in logistics (Konstantakopoulos et al., 2022), transportation (Garaix et al., 2010), and supply chain management (Dondo et al., 2011). Efficiently solving VRPs is critical for reducing operational costs and enhancing service quality in practice. Traditionally, VRPs have been tackled either using exact solvers (e.g., Gurobi) or heuristic solvers (e.g., LKH-3 (Helsgaun, 2017)). While these methods can yield high-quality (or even optimal) solutions for small to moderate-sized instances, they often face challenges in scaling to larger problem sizes or adapting to different problem variants without extensive domain expertise or manual tuning.

Neural solvers have emerged as a promising alternative by leveraging advanced deep learning techniques to learn solution strategies directly from data (Bengio et al., 2021). Numerous neural solvers have been proposed for solving VRPs (Bogyrbayeva et al., 2024), with autoregressive construction solvers gaining particular popularity. These solvers sequentially build solutions by adding one feasible node at a time and are valued for their conceptual simplicity and flexibility across different VRP variants. Among them, (heavy encoder) light decoder-based solvers (Vinyals et al., 2015; Kool et al., 2019; Kwon et al., 2020; Kim et al., 2022; Gao et al., 2024; Liu et al., 2024) stand out for their computational efficiency and ease of integration with reinforcement learning (RL) algorithms. These methods typically employ a heavy transformer-based encoder to compute *static* node embeddings, followed by a lightweight decoder to construct the solution by sequentially selecting the next

---

*Equal contribution.

†Corresponding author.

node based on these embeddings. While this paradigm has shown promising in-distribution performance, it faces significant challenges in generalizing to out-of-distribution (OOD) instances (Joshi et al., 2021), especially those with larger problem sizes or more complex constraints.

Recent works (Drakulic et al., 2023; Luo et al., 2023) propose a (light encoder) heavy decoder architecture to generate *dynamic* node embeddings, improving generalization on large-scale instances. Despite the impressive performance, they exhibit several limitations: 1) their solution construction process is computationally inefficient, as they must re-embed the remaining nodes at each decoding step, resulting in much higher computational costs than light decoder-based solvers; 2) they require a considerable amount of high-quality solutions as labels for supervised training, which may be impractical for less-explored problem variants. Although self-improvement learning (Luo et al., 2025) can help reduce the label burden, advanced search techniques (Pirnay & Grimm, 2024) may be necessary to generate high-quality pseudo-labels for effective training, introducing an extra layer of complexity to the training process; 3) their versability may be compromised by the policy formulation, as they have to solve problems with an inherently tail-recursive nature. That said, both paradigms have their respective merits, and our goal is to narrow the performance gap between them.

In this paper, we revisit the *light decoder* paradigm by analyzing its inherent limitations, and identifying potential bottlenecks in current methods that hinder their performance. Firstly, we demonstrate that the architecture design of the light decoder paradigm imposes an inherent challenge on the encoder by assigning it an overly complex learning task. As nearly all embedding transformations are performed in a context-agnostic manner by the encoder, it has to encapsulate all relevant information necessary to address any possible future context of solution construction within a single set of static embeddings, resulting in high information density. As the problem size grows, this task becomes exponentially more complex, as the encoder must anticipate and represent an increasingly vast amount of potential sub-problems and decision paths within fixed-size static embeddings. In contrast, the dynamic node embeddings in the heavy decoder paradigm are only responsible for predicting the next step based on the current state, simplifying the decision-making task but sacrificing computational efficiency. Secondly, despite these inherent challenges of the light decoder paradigm, their heavy encoders are still capable of learning valuable static embeddings that effectively address a broad spectrum of sub-problems. However, our empirical analysis reveals that the overly simplistic decoder struggles to effectively leverage the dense information embedded in static embeddings during solution construction and fails to handle tasks with unseen levels of complexity, thereby limiting the model's ability to generalize to OOD problem instances. This suggests that enhancing the decoder's capacity could unlock the latent potential of existing approaches and potentially alleviate the burden on the encoder. Building on these insights into the light and heavy decoder paradigms, we propose a simple yet efficient method that leverages the strengths of each to compensate their respective weaknesses, significantly improving the performance of light decoder-based solvers.

Our contributions are summarized as follows. 1) We systematically revisit the light decoder paradigm by thoroughly analyzing its inherent challenges and identifying potential bottlenecks in existing methods, with the goal of enhancing their generalization performance. 2) We provide insights into their poor generalization, including the overly complex learning task imposed on the encoder and the inability of the simplistic decoder to effectively leverage the dense information in static embeddings when addressing OOD instances. Based on these insights, we propose ReLD, incorporating simple yet efficient modifications, such as adding identity mapping and a feed-forward layer, to enhance the decoder's capacity. 3) We conduct extensive experiments on cross-size and cross-problem benchmarks, and demonstrate the effectiveness of our method in significantly improving both in-distribution and OOD generalization performance of light decoder-based solvers. Notably, our work narrows the performance gap between light and heavy decoder paradigms, reaffirming the potential of light decoder-based solvers when properly adjusted.

## 2    RETHINKING LIGHT DECODER-BASED SOLVERS

We introduce light decoder-based solvers, where the decoder leverages *static node embeddings*[1] *as keys and values in its attention layers throughout the entire decoding process.* Hereafter, with a slight abuse of terminology, the terms *static embedding* and *light decoder* will be used interchangeably.

---

[1]In this paper, static embedding refers to a set of node embeddings used for all key and value computations in the decoder's attention layers throughout the entire decoding process.

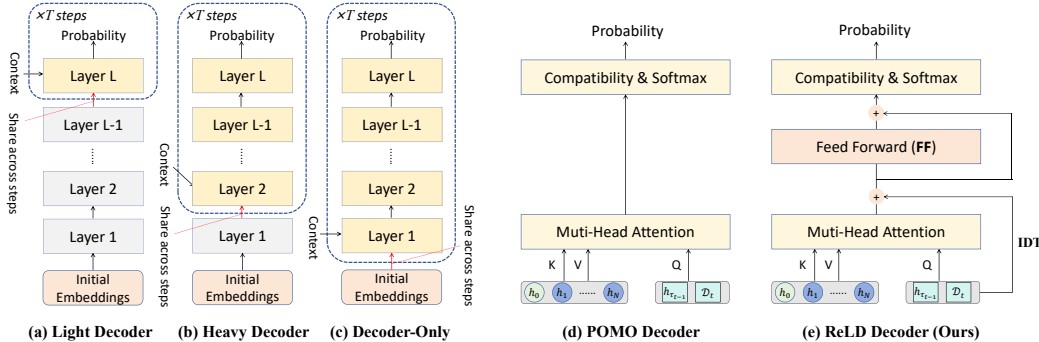

Figure 1: (a)-(c): The primary difference between light decoder, heavy decoder, and decoder-only paradigms lies in the number of prefix layers that are shared across decoding steps. (d)-(e): Decoder structures of POMO and ReLD. QKV in (d) or (e) refers to the query, key and value matrices involve in the computation of MHA as presented in Eq. (4) or (9) (e.g., $Q = h_c$, $K = V = H_t$ in Eq. (4)).

## 2.1 PRELIMINARIES

**VRP.** A VRP instance is defined over a graph $G = \{V, E\}$, where $v_i \in V = \{v_i\}_{i=0}^N$ represents the node and $e(v_i, v_j) \in E$ represents the edge between node $v_i$ and $v_j$. A feasible solution $\tau$ is represented as a sequence of nodes that satisfies the problem-specific constraints. For example, a feasible solution in CVRP consists of multiple sub-tours, each of which represents a vehicle starting from the depot node $v_0$, visiting a subset of customer nodes and returning to the depot node. It is feasible if each customer node is visited exactly once and the total demand in each sub-tour does not exceed the vehicle's capacity limit (i.e., capacity constraint).

**Policy Formulation.** In theory, the Markov decision process (MDP) of light decoder-based solvers for generating a solution $\tau = (\tau_1, \dots, \tau_T)$ to a VRP instance should be formulated as follows:

$$p_\theta(\tau|G) = \prod_{t=2}^T p_\theta(\tau_t|\tau_1, \tau_{t-1}, s_t, U_t), \tag{1}$$

where $p_\theta$ is the policy parameterized by $\theta$, $T$ is the total number of construction steps, $\tau_t$ represents the node selected at step $t$, $s_t$ encapsulates the state variables at step $t$ (e.g., remaining vehicle capacity for CVRP), and $U_t$ denotes the sub-graph of $G$ consisting of only unvisited nodes at step $t$. It is important to note that at any step $t$, the policy should be conditioned only on the immediate previous node $\tau_{t-1}$ and the initial node $\tau_1$ (i.e., depot node $v_0$), rather than the entire sequence of previously selected nodes. This reflects the property that the solution to the remaining sub-problem is independent of the sequence of earlier decisions made in VRPs (Drakulic et al., 2023).

**Light Decoder-based Architecture.** Typically, a transformer-based encoder-decoder architecture is utilized to parameterize the policy $p_\theta$. Given an instance of $N$ customer nodes, each characterized by raw features $x_i \in \mathbb{R}^{d_x}$, the model first projects these features into initial embeddings $h_i^{(0)} \in \mathbb{R}^{d_h}$ through a linear transformation. These initial embeddings are then processed through $L$ encoder layers to generate a final set of static node embeddings $H^{(L)} = (h_0^{(L)}, \dots, h_N^{(L)})$. Formally, each encoder layer $\ell$ transforms $H^{(\ell-1)} = \{h_i^{(\ell-1)}\}_{i=0}^N$ as follows:

$$\hat{h}_i^{(\ell)} = \text{NORM}\left(h_i^{(\ell-1)} + \text{MHA}(h_i^{(\ell-1)})\right), \tag{2}$$

$$h_i^{(\ell)} = \text{NORM}\left(\hat{h}_i^{(\ell)} + \text{FF}(\hat{h}_i^{(\ell)})\right), \tag{3}$$

where $\text{NORM}(\cdot)$ is a normalization layer, $\text{MHA}(\cdot)$ is a multi-head self-attention layer, and $\text{FF}(\cdot)$ is a feed forward network with ReLU activation. We refer to Kool et al. (2019) for further details. At each decoding step $t$, the decoder takes as inputs the node embeddings $H_t = \{h_i^{(L)}\}_{i \in F_t}$, with $F_t$ denoting the set of feasible nodes at time $t$, and a context vector $h_c = [h_{\tau_{t-1}}^{(L)}, \mathcal{D}_t] \in \mathbb{R}^{d_h + d_{attr}}$, with $\mathcal{D}_t \in \mathbb{R}^{d_{attr}}$ denoting dynamic features that capture the state variable $s_t$ in Eq. (1). To aggregate

information from node embeddings, the decoder refines the context vector $h_c$ through a multi-head cross-attention layer, using $h_c$ as the query and $H_t \in \mathbb{R}^{|F_t| \times d_h}$ as the key and value, as follows:

$$h_c' = \text{MHA}(h_c, H_t, H_t). \tag{4}$$

Finally, with a hyperparameter to clip the logits so as to benefit the policy exploration, the probability $p_i$ of selecting node $i \in F_t$ is calculated as follows:

$$p_i = \text{Softmax} \left( C \cdot \tanh(\frac{(h_c')^T H_t}{\sqrt{d_h}}) \right)_i. \tag{5}$$

**Multi-trajectory Strategy.** The multi-trajectory strategy has been a default technique for light decoder-based solvers during training and inference since its introduction in Kwon et al. (2020). Specifically, it exploits the symmetries in VRP solution representations by sampling multiple trajectories of an instance, each starting from a different initial node. This approach can be efficiently implemented due to the shared embedding design in light decoder-based solvers, where all trajectories share the same set of static node embeddings, significantly reducing memory consumption. During training, the model parameters $\theta$ are optimized to minimize the expected tour length over the training data distribution using the REINFORCE algorithm (Williams, 1992). For each instance, multiple trajectories are generated by varying the starting move, with the baseline value for the policy gradient set to the mean reward (i.e., negative tour length) of all trajectories sampled from that instance. At inference time, multiple trajectories are sampled in parallel, and the one with the minimal cost is selected as the final solution for a given test instance. This strategy effectively enhances exploration of the solution space without incurring significant computational overhead.

## 2.2 POLICY REFORMULATION

Light decoder-based solvers are characterized by their reliance on *static embeddings*, where the input node representations for computing keys and values in the decoder's attention layers remain unchanged throughout all decoding steps. Most existing approaches reformulate the decoding policy (as in Eq. (1)) in terms of the encoder $f_{\theta_E}$ and the decoder $g_{\theta_D}$ as follows:

$$p_{\theta_E, \theta_D}(\tau|G) = \prod_{t=2}^{T} g_{\theta_D}(\tau_t|h_{\tau_{t-1}}, s_t, \{h_j\}_{j \in U_t}), \tag{6}$$

where $\{h_i\}_{i=0}^N = f_{\theta_E}(G)$ represents the set of node embeddings for the entire graph $G$ computed by the encoder $f_{\theta_E}$. In light decoder-based methods, the decoder $g_{\theta_D}$ comprises a shallow network, such as a MHA and single-query attention layer (Kool et al., 2019; Kwon et al., 2020), while the encoder $f_{\theta_E}$ is generally a deeper network. In contrast, heavy decoder-based methods feature a deeper decoder paired with a shallow encoder (Luo et al., 2023). Furthermore, designing $f_{\theta_E}$ as either an identity mapping or a simple node-wise feed-forward network can approach what are essentially decoder-only solvers (Drakulic et al., 2023). We refer to Appendix B for a more comprehensive discussion on the relationship between these paradigms within a unified framework.

**Gap between Policy Formulations.** Unfortunately, there is a gap between the practical policy formulation in Eq. (6) and the principled policy formulation in Eq. (1). Concretely, each node embedding $h_j$, used for decision-making at each decoding step, inherently contains information about all other (potentially irrelevant) nodes in the graph due to the attention mechanisms in the encoder. Consequently, at each step, the decoder in Eq. (6) is implicitly conditioned on all previously selected nodes, which diverges from the principled policy formulation. This misalignment exhibits the gap between the theory and practical implementations in existing neural solvers. In the following, we explore this discrepancy from an alternative perspective.

**Static Embedding as KV Cache.** Recent advancements in Large Language Models (LLM) involve using KV Cache techniques to store computed keys and values from previous tokens within attention layers in the decoder, reducing computational time (Shazeer, 2019; Brown, 2020; Ainslie et al., 2023; Touvron et al., 2023). Static embedding in COP shares similar motivation with KV Cache in NLP, aiming to prevent the re-computation of keys and values. However, a fundamental difference exists in the MDP formulation between COP and NLP. In NLP, the process is *generative*: each predicted token is appended to the state (or context) for subsequent predictions, i.e., the context is

increasing throughout the decoding process. Caching KV aligns naturally with the generative nature of the task, where each new token depends on the extended historical context, and hence it does not disrupt the original MDP formulation of the task. In contrast, the process in COP is *selective*: the context diminishes as each predicted element (or token) is removed from consideration in subsequent decisions. This reduction in context, contrary to the expansion observed in NLP, implies that *simply reusing KV computed on historical context may introduce irrelevant information to the decision on current sub-problem*. This increases the learning task's complexity, as the decoder further needs to learn to disentangle irrelevant information from the inputs.

### 2.3 ANALYSES OF LIGHT DECODER PARADIGM

In the light decoder paradigm, the encoder takes on the complex task of producing context-agnostic node embeddings that are broadly applicable across various sub-problems within the original instance, yielding static embeddings with high information density. Despite the challenging nature of the task, we find that these embeddings are effective in addressing a broad spectrum of sub-problems. Then, the decoder makes decisions based on the current sub-problem at hand to determine the next step. However, our empirical analysis reveals that the simplistic decoder struggles to effectively leverage the dense information in static embeddings and fails to manage tasks with unseen levels of complexity, thereby limiting the model's ability to generalize to OOD problem instances. This highlights the need for a more powerful decoder capable of effectively interpreting and adapting static embeddings to the current context, filtering out irrelevant information and focusing on the features most relevant to the sub-problem at hand. In the following, we delve into each point in detail.

**Complex Learning Task for the Encoder.** A weak decoder imposes substantial challenges on the encoder's learning task. Since 1) nearly all embedding transformations are performed in a context-agnostic manner by the encoder, and 2) the simplistic decoder lacks the capacity to effectively adjust static embeddings according to the context, the encoder has to encapsulate all relevant information to produce static embeddings that are sufficiently detailed and informative. These static embeddings are expected to handle any potential sub-problem (i.e. context) that may arise during solution construction, resulting in high information density within a single set of embeddings. As the problem size increases, the task becomes exponentially more difficult due to the combinatorial explosion of decision space. Intuitively, this exponential increase in complexity suggests that the optimal strategies for solving problems at different scales may diverge significantly. As the problem size grows, the interactions between nodes become more intricate, and additional constraints or patterns that are not prominent in smaller instances may emerge. Therefore, the strategies learned at smaller scales may not generalize well to larger instances, hindering the model's ability to adapt across various problem sizes. Additionally, since the encoder must allocate its representational capacity across a wide range of possible contexts, the quality of embeddings for individual situations may be diluted. This dilution may diminish the encoder's ability to capture specific features necessary for optimal performance. In contrast, in the heavy decoder paradigm, the model only needs to produce embeddings tailored to predict the next step based on the current sub-problem, which is a much simpler task compared to the encoder's burden in the light decoder paradigm. By decomposing the complex task into predicting one step at a time based on dynamic embeddings, heavy decoder-based solvers avoid the exponentially increasing task complexity associated with larger problem sizes.

**Capability of Static Embeddings in Solving Sub-problems.** Despite the challenging nature of the encoder task in light decoder-based solvers, we find that the embeddings generated by heavy encoders are effective in addressing a broad spectrum of sub-problems. To empirically support this, we conduct an experiment using node embeddings obtained from CVRP instances to solve randomly sampled sub-instances. For simplicity, instead of generating a large instance and subsampling a set of nodes, we first generate a small instance (i.e., a subproblem) and then add a random set of additional nodes. Note that these two processes are equivalent, as the generation of each node is independent. Specifically, for a uniformly sampled CVRP instance of size $n$, we introduce an additional set of irrelevant nodes of size $\lceil \delta \times n \rceil$ generated from the same distribution, where $\delta$ denotes the extension rate. Then, the encoder processes all $n + \lceil \delta \times n \rceil$ nodes, but only the original $n$ node embeddings are used as inputs to the decoder for decision-making. We conduct experiments on 100 instances and report the average results in Table 1. It can be observed that using embeddings from a larger instance does not significantly affect the performance of light decoder-based solvers, such as POMO (Kwon et al., 2020). Since the model is unaware of the specific sub-problem it needs to solve in advance, this suggests that static embeddings contain valuable information to solve

Table 1: Impact of embeddings from an extended graph. Models are trained on CVRP100.

| Method | $\delta$ | CVRP100 Gap | CVRP200 Gap | CVRP500 Gap |
|---|---|---|---|---|
| POMO | 0.0 | 1.000% | 3.403% | 11.135% |
|  | 0.5 | 1.556% | 3.968% | 10.117% |
|  | 1.0 | 1.880% | 4.299% | 10.255% |
| AM | 0 | 7.094% | 10.707% | 18.145% |
|  | 0.5 | 7.375% | 10.729% | 20.129% |
|  | 1.0 | 7.642% | 11.307% | 23.746% |
| LEHD | 0.0 | 3.648% | 3.312% | 3.178% |
|  | 0.5 | 11.449% | 12.450% | 9.999% |
|  | 1.0 | 12.951% | 14.456% | 12.123% |
| BQ | 0.0 | 2.726% | 2.972% | 3.248% |
|  | 0.5 | 12.772% | 13.647% | 11.238% |
|  | 1.0 | 15.993% | 17.504% | 13.965% |

Table 2: Experiments on fine-tuning and model complexity. Models are trained on CVRP100.

| Method | CVRP100 Gap | CVRP200 Gap | CVRP500 Gap | CVRP1000 Gap |
|---|---|---|---|---|
| POMO | 1.000% | 3.403% | 11.135% | 110.632% |
| Fine-tune Dec | 2.620% | 4.487% | 7.148% | 15.202% |
| Fine-tune Enc | 3.061% | 4.384% | 4.863% | 9.627% |
| Fine-tune All | 2.594% | 4.132% | 5.718% | 11.708% |
| POMON | 0.947% | 3.864% | 13.008% | 33.262% |
| POMON-Enc$^+$ | 0.890% | 2.703% | 9.578% | 30.198% |
| POMON-Dec$^+$ | 0.682% | 2.640% | 8.185% | 18.076% |

various sub-problems. Moreover, we conduct similar experiments on heavy decoder-based solvers, such as LEHD (Luo et al., 2023). In this setup, at each decoding step, we append $\lceil \delta \times n \rceil$ random nodes to the model and filter them out before the final decoder layer, ensuring that the additional nodes pass through the same number of layers as in POMO's encoder. The results reveal that the LEHD model is highly sensitive to the additional context, indicating that most of its layers, unlike those in POMO, are highly specialized for solving a specific sub-problem at each decoding step.

**Inability of Simplistic Decoders.** Our empirical results (in Table 2) highlight the inferior generalization performance of light decoder-based solvers on large-scale instances. To further investigate the influence of the encoder and decoder on generalization, we conduct two experiments. First, inspired by the linear probe protocol in self-supervised learning (He et al., 2020; Chen et al., 2020), we fine-tune POMO (pretrained on CVRP100) on large-scale instances. We observe that fine-tuning only the decoder results in relatively poor performance compared to fine-tuning the encoder or performing full fine-tuning. This suggests a potential defect of the decoder architecture, which may complicate the fine-tuning process. Second, we attempt to strengthen the encoder and decoder of POMON (without the normalization layer), respectively. For example, POMON-Enc$^+$ simply incorporates two more layers into the encoder. The results suggest that the decoder may be overly simplistic, as increasing its capability can significantly improve its generalization performance and may reduce the burden on the encoder. This observation is also supported by recent studies (Drakulic et al., 2023; Luo et al., 2023; Zhou et al., 2024b), which show that enhancing the decoder with deeper layers or conditional computation can substantially boost generalization performance. Therefore, we assume the potential bottleneck of light decoder-based solvers lies in their simplistic decoders, which struggle to effectively leverage the dense information in static embeddings and fail to manage tasks with increasing complexity.

## 3 METHODOLOGY

Building on the insights into the light and heavy decoder paradigms discussed in Section 2, we propose ReLD, which incorporates simple yet efficient modifications to the decoder architecture and training process. ReLD effectively enhances the decoder's capability while preserving the advantages of light decoder-based solvers. The structure of ReLD is illustrated in Fig. 1.

### 3.1 DIRECT INFLUENCE OF CONTEXT

Existing light decoder-based solvers typically employ a decoder architecture composed of an MHA layer followed by a compatibility layer. The MHA layer aggregates information from the static node embeddings using a context vector $h_c$ to form the query vector $h'_c$. This query vector is then used by the compatibility layer to compute the probability distribution for the next step, based on the dot-product similarity between $h'_c$ and each node embedding. In this design, $h'_c$ is the only component that captures the context, placing the entire burden of context representation on this single vector. To make the computation of the query vector $h'_c$ clearer, we re-write Eq. (4) as follows:

$$h'_c = \text{MHA}(h_c, H_t, H_t) = \sum_{s=1}^{S} W_s^O \sum_{i \in F_t} u_{s,i} W_s^V h_i = \sum_{s=1}^{S} \sum_{i \in F_t} u_{s,i} v_{s,i}, \qquad (7)$$

$$\text{where } u_{s,i} = \text{Softmax}\left(\frac{(W_s^Q h_c)^T W_s^K H_t}{\sqrt{d_h}}\right)_i, \; v_{s,i} = W_s^O W_s^V h_i. \tag{8}$$

Here, $S$ is the number of attention heads, $W_s^O, W_s^Q, W_s^K, W_s^V$ are the projection matrices in the $s$-th attention head, and $H_t = \{h_i\}_{i \in F_t}$ denotes the static node embeddings. We omit the superscript $(L)$ for simplicity of notation. This design implies that the influence of the context $h_c$ on $h_c'$ is only mediated through the attention weights $u_{s,i}$, without directly modifying the combined vectors. Such an indirect incorporation of context may lead to inefficient utilization of context information, especially in tasks with complex constraints, where dynamic context is crucial for decision-making.

To address this limitation, we propose directly integrating context information into the embedding space, rather than relying solely on the attention weights. Specifically, we suggest adding an residual connection (He et al., 2016) between the context vector $h_c$ and the query vector $h_c'$, thereby embedding context-aware information directly into the representation:

$$h_c' = \text{MHA}(h_c, H_t, H_t) + \text{IDT}(h_c). \tag{9}$$

In practice, $h_c = [h_{\tau_{t-1}}, \mathcal{D}_t]$ typically has a different shape compared to $h_c'$, so the identity mapping function $\text{IDT}(\cdot)$ must reshape the original input. To effectively preserve the information of the last node location $h_{last}$ while maintaining simplicity, we define the function $\text{IDT}(\cdot)$ as follows:

$$\text{IDT}(h_c) = h_{\tau_{t-1}} + W^{\text{IDT}} \mathcal{D}_t, \tag{10}$$

where $W^{\text{IDT}}$ is a learnable parameter that projects $\mathcal{D}_t$ to the same dimension as $h_{\tau_{t-1}}$.

## 3.2 POWERFUL QUERY

A key component of the decoder is the context-aware query vector $h_c'$, which is used to compute attention scores over the node embeddings and ultimately determine the probability distribution over the next possible nodes. This query vector is crucial, as it is the only element in the computation that incorporates the current context of the decision-making. In existing implementations, as shown in Eq. (7), $h_c'$ is computed as a weighted combination of the node embeddings. The computation follows a largely linear form with respect to the fixed sets of value vectors $v_{s,i}$, with the only non-linearity introduced through the attention weights $u_{s,i}$. This inherent linearity may limit the decoder's capacity to model complex relationships and adapt the embeddings based on the context.

Considering the importance of non-linearity in enabling neural networks to model complex functions (Cybenko, 1989), we propose to enhance the decoder's representation capacity by introducing non-linearity into the computation of $h_c'$. Intuitively, to more effectively transform the context information, non-linearity should be introduced after the context has been fully aggregated. This suggests that adding non-linearity after the MHA layer, where the global context is integrated, would have a bigger impact. Drawing inspiration from the observation that the overall architecture of transformer blocks, rather than solely the self-attention mechanism, plays a significant role in the model's performance (Raghu et al., 2021; Yu et al., 2022), we propose incorporating a feed-forward network with residual connections into the decoder as follows:

$$q_c = h_c' + \text{FF}(h_c') = h_c' + \sigma(W_2 \sigma(W_1 h_c' + b_1) + b_2), \tag{11}$$

where $W_1, W_2, b_1$, and $b_2$ are the parameters of the feed-forward layer, and $\sigma$ denotes the ReLU activation function. This modification, along with the one presented in Eq. (9), transforms the decoder into a transformer block with a single query, adding non-linear processing capabilities to the query vector computation. In doing so, it provides the encoder with greater flexibility in designing the embedding space and potentially alleviates the burden imposed on it. Note that our modification is computationally efficient, as the additional step-wise running time is independent of the number of nodes, and therefore does not affect the overall asymptotic time complexity of the decoder. This makes it a practical and effective enhancement for light decoder-based methods.

## 3.3 GENERALIZABLE TRAINING

**Distance Heuristic.** Recent advancements have incorporated distance heuristics into the learned policy (Jin et al., 2023; Gao et al., 2024; Wang et al., 2024; Zhou et al., 2024a) to enhance generalization on larger-scale instances. Building on this idea, we add a negative logarithmic distance

Table 3: Results on synthetic CVRP instances.

| Method | CVRP100 Gap | Time | CVRP200 Gap | Time | CVRP500 Gap | Time | CVRP1000 Gap | Time |
|---|---|---|---|---|---|---|---|---|
| LKH3 | * | 12 h | * | 2.1 h | * | 5.5 h | * | 7.1h |
| HGS | -0.533% | 4.5 h | -1.126% | 1.4 h | -1.794% | 4 h | -2.162% | 5.3 h |
| GLOP-G (LKH3)* | - | - | - | - | - | - | 6.903% | 1.7m |
| BQ greedy* | 2.726% | 1.3 m | 2.972% | 10 s | 3.248% | 0.69m | 5.892% | 1.59m |
| LEHD greedy | 3.648% | 0.38m | 3.312% | 2 s | 3.178% | 15s | **4.912%** | 1.34m |
| MDAM bs50* | 2.211% | 25m | 4.304% | 3m | 10.498% | 12m | 27.814% | 47m |
| POMO augx8 | 1.004% | 1.07m | 3.403% | 2s | 11.135% | 13s | 110.632% | 0.88m |
| ELG augx8 | 1.207% | 3.18 m | 2.553% | 7s | 5.472% | 0.44m | 10.760% | 1.42m |
| ReLD augx8 | **0.960%** | 1.34m | **1.654%** | 2s | **2.975%** | 15s | 6.757% | 0.91m |

Table 4: Results on CVRPLib Set-X instances.

| | $N \leq 200$ (22 instances) | $200 < N \leq 500$ (46 instances) | $500 < N \leq 1000$ (32 instances) | Total (100 instances) |
|---|---|---|---|---|
| BKS | * | * | * | * |
| LEHD greedy* | 11.35% | 9.45% | 17.74% | 12.52% |
| BQ greedy* | - | - | - | 9.94% |
| INViT-3V* | 6.52% | 9.11% | 10.21% | - |
| POMO | 5.63% | 9.25% | 18.85% | 11.52% |
| ICAM no-aug* | 5.14% | 4.44% | 5.17% | 4.83% |
| ELG no-aug | 6.25% | 7.58% | 10.02% | 8.07% |
| ELG | 4.51% | 5.52% | 7.81% | 6.03% |
| MTL | 9.49% | 10.96% | 16.80% | 12.51% |
| MVMoE | 4.74% | 9.97% | 19.60% | 11.90% |
| ReLD-MTL | 3.31% | 5.84% | 9.05% | 6.31% |
| ReLD-MoEL | 3.14% | 5.75% | 8.83% | 6.16% |
| ReLD-MoEL$^+$ | **2.41%** | **3.40%** | **4.52%** | **3.54%** |
| ReLD no-aug | 3.36% | 3.93% | 5.12% | 4.18% |
| ReLD | 2.53% | 3.44% | 4.64% | 3.62% |

heuristic score to the output of the compatibility layer in the decoder, modifying Eq. (5) into:

$$p_i = \text{Softmax} \left( C \cdot \tanh(\frac{(q_c)^T H_t}{\sqrt{d_h}} - \log(dist_i)) \right)_i, \tag{12}$$

where $dist_i$ denotes the distance between node $i$ and the last selected node $\tau_{t-1}$.

**Varying Attribute.** Recent works have also introduced several strategies to improve the generalization on large-scale instances, such as training on varying instance sizes (Drakulic et al., 2023; Zhou et al., 2023; Luo et al., 2023; Wang et al., 2024; Zhou et al., 2024a) and vehicle capacities (Gao et al., 2024). We also adopt these strategies in our approach (see Section 4.1).

## 4 EXPERIMENTAL RESULTS AND ANALYSIS

In this section, we present empirical results from experiments on synthetic CVRP instances of various scales, multi-task cross-problem learning across 16 VRP variants, and the CVRPLib benchmark datasets to demonstrate the effectiveness of the proposed ReLD method. All experiments were conducted using a single Nvidia A800 GPU with 80GB of memory.

**Baselines.** 1) *Traditional Solvers*: We utilize SOTA non-learning solvers LKH3 (Helsgaun, 2017) and HGS (Vidal, 2022), which are known for providing strong results on VRPs. In line with previous work (Luo et al., 2023), the performance gaps are computed relative to LKH3, even though HGS generally produces slightly better results. 2) *Learning-based Solvers*: We retrain the POMO model (Kwon et al., 2020) for 5,000 epochs, with 10,000 instances per epoch and batch size of 64. Additionally, we utilize the publicly released available models MDAM (Xin et al., 2021a), BQ (Drakulic et al., 2023), ELG (Gao et al., 2024), LEHD (Luo et al., 2023), and GLOP (Ye et al., 2024). Results for INViT (Fang et al., 2024) and ICAM (Zhou et al., 2024a) are obtained from their original papers.

### 4.1 CROSS-SIZE

**Problem Setting.** We first evaluate performance on the CVRP using the test dataset introduced in (Luo et al., 2023), which consists of 10,000 instances with 100 nodes, and 128 instances for each problem size of 200, 500, and 1,000 nodes. The corresponding vehicle capacity settings for these problem scales are 50, 80, 100, and 250, respectively.

**Model Setting.** Based on POMO (Kwon et al., 2020), we remove normalization layers in the encoder and add identity mapping (Section 3.1), feed forward layer (Section 3.2) and distance score (Section 3.3) to decoder to obtain ReLD.

**Training Setting.** ReLD employs the same training algorithm as POMO (Kwon et al., 2020), but is trained over 90 epochs, with each epoch comprising 600,000 training instances and a batch size of 120. We use the Adam optimizer with an initial learning rate of 1e-4, which is decayed by a factor of 0.1 at the 70th and 80th epochs. Weight decay is set to zero. For generating the training data, we first sample a problem size from a discrete uniform distribution, Uniform(40, 100). Following the standard data generation procedure outlined in (Kool et al., 2019), node coordinates and demands are generated uniformly. To ensure adaptability to varying attribute (Section 3.3), we randomly sample an expected route size $r$ (average number of nodes in a route) from a triangular distribution

Table 5: Performance on 1K test instances of 16 VRP variants.

| Method | CVRP Obj | Gap | Time | VRPTW Obj | Gap | Time | OVRP Obj | Gap | Time | VRPL Obj | Gap | Time |
|---|---|---|---|---|---|---|---|---|---|---|---|---|
| HGS | 15.504 | * | 9.1m | 24.339 | * | 19.6m | - | - | - | 16.496 | - | - |
| LKH3 | 15.590 | 0.556% | 18.0m | 24.721 | 1.584% | 7.8m | 10.010 | * | 5.3m | 16.004 | * | 16.0m |
| OR-Tools | 15.935 | 2.751% | 3.5h | 25.212 | 3.482% | 3.5h | 9.842 | 0.122% | 3.5h | 15.785 | 1.444% | 3.5h |
| POMO | 15.734 | 1.488% | 9s | **25.367** | **4.307%** | 11s | **10.044** | **2.192%** | 8s | 15.785 | 0.093% | 9s |
| POMO-MTL | 15.790 | 1.846% | 9s | 25.610 | 5.313% | 11s | 10.169 | 3.458% | 8s | 15.846 | 0.479% | 9s |
| MVMoE-L | 15.771 | 1.728% | 10s | 25.519 | 4.927% | 11s | 10.145 | 3.214% | 9s | 15.821 | 0.323% | 10s |
| MVMoE | 15.760 | 1.653% | 11s | 25.512 | 4.903% | 12s | 10.138 | 3.136% | 10s | 15.812 | 0.261% | 11s |
| ReLD-MTL | 15.723 | 1.416% | 10s | 25.432 | 4.561% | 11s | 10.057 | 2.321% | 9s | 15.773 | 0.020% | 10s |
| ReLD-MoEL | **15.713** | **1.354%** | 10s | 25.403 | 4.444% | 11s | 10.050 | 2.260% | 9s | **15.768** | **-0.016%** | 10s |
| ReLD-MoEL$^+$ | 15.739 | 1.515% | 10s | 25.445 | 4.621% | 11s | 10.084 | 2.592% | 9s | 15.788 | 0.113% | 10s |

| Method | VRPB Obj | Gap | Time | OVRPTW Obj | Gap | Time | OVRPB Obj | Gap | Time | OVRPL Obj | Gap | Time |
|---|---|---|---|---|---|---|---|---|---|---|---|---|
| OR-Tools | 11.878 | * | 3.5h | 14.380 | * | 3.5h | 8.365 | * | 3.5h | 9.790 | * | 3.5h |
| POMO | 11.993 | 0.995% | 9s | **14.728** | **2.467%** | 11s | - | - | - | 10.126 | 3.441% | 9s |
| POMO-MTL | 12.072 | 1.674% | 9s | 15.008 | 4.411% | 11s | 8.979 | 7.335% | 8s | 10.106 | 3.244% | 9s |
| MVMoE-L | 12.036 | 1.368% | 10s | 14.940 | 3.941% | 11s | 8.972 | 7.243% | 9s | 10.097 | 3.148% | 10s |
| MVMoE | 12.027 | 1.285% | 11s | 14.927 | 3.852% | 12s | 8.959 | 7.088% | 10s | 10.097 | 3.148% | 11s |
| ReLD-MTL | 11.981 | 0.901% | 10s | 14.818 | 3.103% | 11s | 8.814 | 5.361% | 9s | 10.014 | 2.310% | 10s |
| ReLD-MoEL | **11.969** | **0.801%** | 10s | 14.804 | 3.000% | 11s | 8.804 | 5.239% | 9s | **10.007** | **2.233%** | 10s |
| ReLD-MoEL$^+$ | 11.981 | 0.887% | 10s | 14.848 | 3.304% | 11s | **8.794** | **5.113%** | 9s | 10.040 | 2.567% | 10s |

| Method | VRPBL Obj | Gap | Time | VRPBTW Obj | Gap | Time | VRPLTW Obj | Gap | Time | OVRPBL Obj | Gap | Time |
|---|---|---|---|---|---|---|---|---|---|---|---|---|
| OR-Tools | 11.790 | * | 3.5h | 25.496 | * | 3.5h | 25.195 | * | 3.5h | 8.348 | * | 3.5h |
| POMO-MTL | 11.998 | 1.793% | 9s | 27.319 | 7.413% | 9s | 25.619 | 1.920% | 8s | 8.961 | 7.343% | 9s |
| MVMoE-L | 11.960 | 1.473% | 10s | 27.265 | 7.190% | 10s | 25.529 | 1.545% | 9s | 8.957 | 7.300% | 10s |
| MVMoE | 11.945 | 1.346% | 11s | 27.236 | 7.078% | 12s | 25.514 | 1.471% | 10s | 8.942 | 7.115% | 11s |
| ReLD-MTL | 11.905 | 1.005% | 10s | 27.144 | 6.735% | 11s | 25.433 | 1.167% | 9s | 8.800 | 5.411% | 10s |
| ReLD-MoEL | **11.894** | **0.917%** | 10s | **27.106** | **6.574%** | 11s | **25.415** | **1.089%** | 9s | 8.786 | 5.247% | 10s |
| ReLD-MoEL$^+$ | 11.908 | 1.024% | 10s | 27.166 | 6.806% | 11s | 25.456 | 1.258% | 9s | **8.779** | **5.154%** | 10s |

| Method | OVRPBTW Obj | Gap | Time | OVRPLTW Obj | Gap | Time | VRPBLTW Obj | Gap | Time | OVRPBLTW Obj | Gap | Time |
|---|---|---|---|---|---|---|---|---|---|---|---|---|
| OR-Tools | 14.384 | * | 3.5h | 14.279 | * | 3.5h | 25.342 | * | 3.5h | 14.250 | * | 3.5h |
| POMO-MTL | 15.879 | 10.453% | 9s | 14.896 | 4.374% | 11s | 27.247 | 7.746% | 8s | 15.738 | 10.498% | 9s |
| MVMoE-L | 15.841 | 10.188% | 10s | 14.839 | 3.971% | 11s | 27.177 | 7.473% | 9s | 15.706 | 10.263% | 10s |
| MVMoE | 15.808 | 9.948% | 11s | 14.828 | 3.903% | 12s | 27.142 | 7.332% | 10s | 15.671 | 10.009% | 11s |
| ReLD-MTL | 15.711 | 9.288% | 10s | 14.721 | 3.159% | 11s | **27.042** | 6.937% | 9s | 15.555 | 9.215% | 10s |
| ReLD-MoEL | **15.697** | **9.184%** | 10s | **14.707** | **3.054%** | 11s | 27.044 | **6.915%** | 9s | **15.550** | **9.171%** | 10s |
| ReLD-MoEL$^+$ | 15.728 | 9.403% | 10s | 14.744 | 3.318% | 11s | 27.078 | 7.106% | 9s | 15.599 | 9.516% | 10s |

$T(3, 6, 25)$ (Uchoa et al., 2017) for each instance, the vehicle capacity of which is set to $\lceil r\bar{D} \rceil$, where $\bar{D}$ represents the mean demand of the nodes in the instance.

**Inference Setting.** For all neural solvers, we adopt the greedy rollout strategy. To ensure comparable runtime across baselines, we set the number of rollout trajectories $K$ for all light decoder-based methods (POMO, ELG, ReLD) to $min(100, N)$, where $N$ represents the problem size. To initialize each trajectory, we input all the nodes with full capacity to the decoder and select top $K$ moves with respect to the output probability distribution.

**Results.** The results on uniformly generated CVRP instances are presented in Table 3 (the superscript asterisk (*) on method denotes that the results of the method are directly obtained from the original paper). ReLD consistently outperforms other learning-based solvers at all scales except for CVRP1000, while achieving similar or lower computational time. For CVRP1000, ReLD remains highly competitive, surpassing most neural solvers. However, solvers like BQ and LEHD still demonstrate superior performance at this scale, indicating that a performance gap persists between light decode methods and architectures with more decoder layers for larger instances.

### 4.2 CROSS-PROBLEM

We follow all the problem setting, training setting, and evaluation setting used in Zhou et al. (2024b).

**Model Setting.** We remove all the normalization layers in the encoder and add identity mapping (Section 3.1), feed forward layer (Section 3.2) to the original POMO-MTL and MVMoE-light to obtain the ReLD-MTL and ReLD-MoEL. Notably, original MVMoE-light utilizes an MoE layer to replace the linear layer in the decoder to transform the results obtained from multiple attention heads. For simplicity and fair comparison, we use the original POMO decoder architecture with added identity mapping and feed forward layer in both of the ReLD-MTL and ReLD-MoEL. We also employ the training techniques in Section 3.3 on ReLD-MoEL, denoted as ReLD-MoEL$^+$.

**Results.** The results on all the VRP variants are shown in Table 5 and Table 7 (see Appendix). It is quite clear that all the ReLD models outperform the POMO-MTL, MVMoE-L and MVMoE baselines on all the studied VRP variants by a significant margin. **Notably**, both the ReLD-MTL and ReLD-MoEL outperform the single-task POMO that is specifically trained on the CVRP, VRPL and

VRPB. An even more surprising result is that ReLD-MoEL surpasses the strong LKH3 traditional solver on VRPL, achieving a negative gap. The improvement becomes even more pronounced in the generalization to unseen VRP variants (the last 10 variaints in Table 5), with an average gap improvement of 1.1% and 1.0% for ReLD-MTL and ReLD-MoEL, respectively, compared to their original counterparts. This demonstrates the superior ability of ReLD to handle varying constraints effectively. On the other hand, ReLD-MoEL$^+$ underperforms relative to other ReLD models on most problems, though it still significantly outperforms all baselines. This suggests that the techniques employed in Section 3.3 may trade off in-distribution performance to gain stronger out-of-distribution generalization, aligning with the observations in Section 4.4.

### 4.3 Benchmark Performance

We also evaluate our ReLD on the well known benchmark dataset CVRPLib Set-X (Uchoa et al., 2017) and Set-XXL. Specifically, we divide CVRPLib Set-X (Uchoa et al., 2017) into 3 subsets according to the scale the of instances, each with range $N \leq 200$, $200 < N \leq 500$ and $500 < N \leq 1000$. For Set-XXL, we select instances with size $3000 \leq N \leq 16000$.

**Inference.** On both datasets, light decoder-based models run with a maximum of 100 trajectory and augmentation x8 (Kwon et al., 2020) is used by default.

**Results.** The results on Set-X and Set-XXL are presented in Table 4 and Table 8 (see Appendix), respectively. Interestingly, heavy decoder methods like LEHD show inferior performance on Set-X instances, especially when compared to the lighter decoder-based methods. ReLD models, including both ReLD-MTL and ReLD-MoEL, consistently outperform the baseline methods without distance heuristic across all instance sizes. For the models that involve using advanced techniques like distance heuristic, ReLD-MoEL$^+$ achieves the best results. Such surprising results suggest that it is a generalizable foundational model. Furthermore, ReLD demonstrates a significant margin of improvement over all baseline methods on the four large scale real-world instances from Set-XXL, further validating the generalizability of the proposed methods.

### 4.4 Ablation Study

We conduct an ablation study to gain insights for the contribution of each components in ReLD for the overall performance. The main results are summarized in Tables 10 and 11. We refer readers to Appendix G for further details and analysis.

**Identity Mapping is Crucial for OOD Size Generalization.** The introduction of identity mapping significantly improves generalization to larger problem sizes for both POMO and POMON (POMO without the normalization layer), highlighting its critical role in handling out-of-distribution (OOD) instances. Additionally, it provides minor improvements for smaller problem sizes, indicating that it enhances generalization without sacrificing in-distribution performance.

**Feed Forward Layer is Crucial for In-Distribution Learning.** Adding feed-forward layer to POMON reduces the gap on CVRP100 from 0.95% to 0.66% (POMON v.s. POMON+FF), a significant improvement in in-distribution learning. However, this modification is less effective for larger scales without the identity mapping, as seen on CVRP1000.

## 5 Conclusion

This paper positions light decoder-based models as efficient approximations to the original MDP formulation of VRP by sharing computations across decoding steps. We provide valuable insights into potential bottlenecks of existing light decoder methods and introduce ReLD, a simple yet effective approach that narrows the performance gap between light and heavy decoder paradigms. Extensive experiments on cross-size and cross-problem benchmarks demonstrate the effectiveness of our method. However, ReLD still trails behind heavy decoder-based solvers on very large-scale instances. This highlights an inherent trade-off between performance and computational efficiency. Future work includes exploring the boundaries of light decoder-based solvers and investigating optimal performance-efficiency trade-offs by controlling the amount of computation sharing.

ACKNOWLEDGMENTS

This work is supported by the National Research Foundation, Singapore under its AI Singapore Programme (AISG Award No. AISG3-RP-2022-031), and the Singapore Ministry of Education (MOE) Academic Research Fund (AcRF) Tier 1 grant.

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

## A  RELATED WORK

Neural VRP solvers typically learn construction heuristics, which can be broadly categorized into two types: autoregressive and non-autoregressive construction solvers. Autoregressive construction solvers build solutions sequentially by adding one feasible node at a time. Within this category, *light decoder-based solvers*, characterized by static node embeddings generated in a heavy encoder and solution construction handled by a lightweight decoder, have gained significant popularity for their computational efficiency and versatility. Vinyals et al. (2015) propose the Pointer Network, which solves the traveling salesman problem (TSP) using supervised learning (SL). Later works shift from SL to RL to solve TSP (Bello et al., 2017) and the capacitated vehicle routing problem (CVRP) (Nazari et al., 2018). Kool et al. (2019) introduce the attention model (AM), targeting a range of COPs such as TSP and CVRP. Building on this, Kwon et al. (2020) propose the policy optimization with multiple optima (POMO), which improves AM by exploiting a multi-trajectory strategy. Subsequent developments (Kwon et al., 2021; Li et al., 2021a; Kim et al., 2022; Grinsztajn et al., 2023; Chen et al., 2023; Liu et al., 2024; Gao et al., 2024; Zhou et al., 2024b; Berto et al., 2023; 2024; Bi et al., 2024; Hottung et al., 2025; Hua et al., 2025) are often built upon AM and POMO, further advancing their scalability (Li et al., 2021b; Hou et al., 2023; Ye et al., 2024) and generalization (Joshi et al., 2021; Bi et al., 2022; Geisler et al., 2022; Zhou et al., 2023). *Heavy decoder-based solvers* (Drakulic et al., 2023; Luo et al., 2023; 2025), characterized by dynamic node embeddings re-embedded at each decoding step, demonstrate impressive performance on large-scale instances, though at the expense of computational efficiency and training simplicity. Regarding non-autoregressive construction solvers, solutions are generated in a one-shot manner, bypassing the need for iterative forward passes through the model. Early work (Joshi et al., 2019) employs a graph convolutional network to predict the probability of each edge appearing on the optimal tour (i.e., a heatmap) using SL. More recent efforts (Fu et al., 2021; Qiu et al., 2022; Sun & Yang, 2023; Min et al., 2023; Ye et al., 2023; Kim et al., 2024; Xia et al., 2024) further enhance performance by leveraging advanced models, training paradigms, and search strategies. Beyond construction solvers, some neural solvers learn improvement heuristics to iteratively refine an initial feasible solution until a termination condition is met. These approaches often involve learning more efficient or effective search components within classical local search methods or specialized heuristic solvers (Chen & Tian, 2019; Lu et al., 2020; Costa et al., 2020; Wu et al., 2021; Ma et al., 2021; Xin et al., 2021b; Ma et al., 2023). While improvement solvers generally outperform construction solvers in terms of solution quality, they come with the trade-off of significantly longer inference time. In this paper, we mainly focus on autoregressive construction solvers due to their efficiency and versatility in solving VRPs. Moreover, extensive research has focused on solving other COPs, such as the scheduling problem (Zhang et al., 2020; Smit et al., 2024) and graph COPs (Zhang et al., 2023). We refer readers to Bengio et al. (2021); Cappart et al. (2023) for a comprehensive survey.

## B  MORE DISCUSSIONS ON LIGHT DECODER AND HEAVY DECODER

While the main text focuses on the differences in encoder and decoder depth across paradigms, an alternative and unifying perspective is to view these architectures through the lens of *layer sharing across decoding steps*. Specifically, for a model with $L$ layers, the computation of the first $L'$ layers are shared across $K$ decoding steps. This general framework captures existing methods as special cases: **decoder-only** models (e.g., BQ (Drakulic et al., 2023)) arise when $L' = 0$, **heavy decoder** models (e.g., LEHD (Luo et al., 2023)) correspond to cases where $L' < L - 1$, and **light decoder** models (e.g., AM (Kool et al., 2019), POMO (Kwon et al., 2020)) are obtained when $L' = L - 1$. This flexible paradigm enables trade-offs between performance and efficiency by adjusting the number of shared layers and the frequency of updating shared embeddings. Under this framework, static embeddings are utilized for only a constant number of decoding steps K, meaning they only need to encode information relevant to a bounded span of future contexts rather than for all $O(N)$ steps. This reduction in required information density by an order of magnitude alleviates the complexity of the encoder's learning task and, in principle, should lead to improved performance. We leave a comprehensive investigation of this framework for future work.

## C  IMPACT OF CAPACITY

We found that the POMON+IDT+FF model, when trained with a fixed capacity setting, struggles to generalize to instances with capacity distributions that differ significantly from the training distribution, even on relatively small problem size (see Table 6). This suggests that *capacity constraints may introduce generalization challenges that are independent of problem scale*. Motivated by this observation, we explore training with varying capacity settings to expose the model to a broader range of contexts, aiming to improve its adaptability to diverse instance distributions.

Table 6: Investigation into the impact of capacity settings. The POMON+IDT+FF model, trained on instances of size sampled from Uniform(40, 100) with a fixed capacity of 50, struggles to generalize to varying capacity settings. "No-aug single" denotes inference using a single trajectory without augmentation. "n100:Cap$M$" represents instances of size 100 with capacity $M$.

| Method | n100:Cap50 Obj. | n100:Cap100 Obj. | n100:Cap250 Obj. | n100:Cap500 Obj. |
|---|---|---|---|---|
| POMON+IDT+FF no-aug single | 16.20 | 11.14 | 9.04 | 8.65 |
| LEHD greedy | 16.55 | 11.17 | 8.81 | 8.41 |

## D  AVERAGE GAP IN CROSS-PROBLEM SETTINGS

Table 7: Average gap of each model in cross-problem settings.

| Method | 6 Trained VRPs | 10 Unseen VRPs | ALL |
|---|---|---|---|
| POMO-MTL | 2.863% | 6.231% | 4.968% |
| MVMoE-L | 2.584% | 5.989% | 4.712% |
| MVMoE | 2.515% | 5.844% | 4.596% |
| ReLD-MTL | 2.054% | 5.059% | 3.932% |
| ReLD-MoEL | **1.974%** | **4.962%** | **3.842%** |
| ReLD-MoEL$^+$ | 2.172% | 5.126% | 4.019% |

## E  RESULTS ON CVRPLIB SET-XXL

Table 8: Empirical results on real-world instances from CVRPLib Set-XXL with $N \leq 16,000$. The asterisk (*) denotes the results of methods directly obtained from the original paper.

| | L1 (3K) | L2 (4K) | A1 (6K) | A2 (7K) | G1 (10K) | G2 (11K) | B1 (15K) | B2 (16K) |
|---|---|---|---|---|---|---|---|---|
| GLOP-LKH3* | 16.60% | 21.10% | 19.30% | 19.40% | 18.30% | 18.10% | 27.50% | 20.10% |
| BQ greedy* | 13.27% | 24.00% | 11.21% | 15.02% | - | - | - | - |
| LEHD greedy* | 14.04% | 26.30% | 18.90% | 26.40% | 27.23% | 38.45% | 35.94% | 40.76% |
| POMO* | 75.30% | 78.16% | 112.27% | 159.22% | - | - | - | - |
| Sym-POMO* | 170.44% | 179.55% | 79.72% | 179.55% | - | - | - | - |
| Omni-POMO* | 22.79% | 60.39% | 42.52% | 48.59% | - | - | - | - |
| ELG* | 10.77% | 21.80% | 10.70% | 17.69% | 13.24% | 18.93% | 17.73% | 20.05% |
| ReLD no-aug | 7.74% | 15.23% | 7.61% | 13.53% | 7.92% | 15.44% | 10.45% | 15.59% |
| ReLD | **7.45%** | **14.55%** | **7.43%** | **13.50%** | **7.63%** | **14.38%** | **10.08%** | **14.69%** |

## F  APPLICATION ON ATSP

We extend our approach to the Asymmetric Traveling Salesman Problem (ATSP), a more practical variant of TSP. Experiments are conducted based on the MatNet model (Kwon et al., 2021).

### F.1 ENCODER MODIFICATIONS

To create a more efficient and effective encoder, we introduce simple yet impactful modifications to the original MatNet encoder. These modifications are applied to both MatNet and ReLD.

**Initial Embedding.** Original MatNet use zero vectors and one-hot vectors for initial embeddings. However, using one-hot vectors limits the model's applicability to instances with size larger than embedding dimension (256). To resolve this, we assign a random number from Uniform(0, 1) to each node as its initial node features (Drakulic et al., 2025). Then, a learnable linear layer transforms the 1D random features into the embedding dimension as inputs to the encoder.

**Single Branch.** MatNet employs a two-branch transformer architecture with cross-attention to model interactions between two sets of objects. However, this design is unnecessary for ATSP, where only a single set of objects (i.e., nodes) is present. To simplify the architecture while preserving effectiveness, we modify the encoder to use a single-branch self-attention mechanism. This adjustment reduces computational overhead without sacrificing performance.

**Mixed Score Self-Attention.** To modify the mixed score cross-attention proposed in MatNet to self-attention, we mix the distance matrix $D$ and its transpose $D^T$ simultaneously with the attention score of each head. This helps the network better understand the *from-to* dual relationship between nodes with self-attention. Notably, we observed that the model fails to converge when $D^T$ is omitted.

**Normalized Distance Matrix.** Since the optimal solution is invariant to the scale of the distance matrix, we normalize the distance matrix by dividing its maximum value before passing the network.

### F.2 RELD FOR ATSP

We incorporate the proposed architectural enhancements—identity mapping and a feed-forward network—into the MatNet decoder. Specifically, let $h_{\tau_t}$ denote the embedding of the $t^{th}$ node of the constructed trajectory. At step $t + 1$, the query vector $h'_c$ is obtained as follows:

$$h'_c = \text{MHA}([h_{\tau_t}, h_{\tau_1}], H_t, H_t) + h_{\tau_t},$$

where $[\cdot, \cdot]$ denotes the concatenation operator. Notice that we exclude the step-agnostic context information $h_{\tau_1}$ in the identity mapping. To further refine $h'_c$, we apply a feed-forward network with a modified reZero (Bachlechner et al., 2021) connection:

$$q_c = \alpha h'_c + \text{FF}(h'_c),$$

where FF is a 2-layer feed-forward network with ReLU activation, $\alpha$ is a learnable scalar initialized to zero, applied to the identity branch, slightly deviating from the original reZero formulation. The resulting $q_c$ is then used in Eq. (5) of the main paper to output action probabilities. Note that we do not incorporate the distance heuristic or varying-size training techniques in this preliminary result.

### F.3 TRAINING SETTING

We train the models on ATSP instances of size 100 for 2100 epochs, with each epoch consisting of 10,000 instances and a batch size of 128. To reduce memory consumption, we use four attention heads with a query-key-value dimension of 64 in the encoder, while keeping all other hyperparameters identical to those in MatNet (Kwon et al., 2021).

### F.4 RESULTS

We evaluate the models on synthetic ATSP instances of sizes 20, 50, 100, 200, and 500. All models use greedy rollouts with full trajectories and no augmentation. For the original MatNet, we report results from the released checkpoint, which was trained for 12,000 epochs on instances of size 100. MatNet$^+$ incorporates the encoder modifications described in Section F.1, while ReLD builds upon MatNet$^+$ with additional decoder enhancements. The results, presented in Table 9, show that the original MatNet model struggles to generalize across different instance sizes. MatNet$^+$ improves performance over the original model, and ReLD further enhances results across all tested sizes.

Table 9: Results on ATSP instances across different scales.

| Method | ATSP20 Obj. (Gap) | ATSP50 Obj. (Gap) | ATSP100 Obj. (Gap) | ATSP200 Obj. (Gap) | ATSP500 Obj. (Gap) |
|---|---|---|---|---|---|
| CPLEX | 1.540 (*) | 1.560 (*) | 1.570 (*) | - | - |
| MatNet official | 2.608 (69.351%) | 2.556 (63.846%) | 1.624 (3.439%) | - | - |
| MatNet$^+$ | 1.651 (7.208%) | 1.604 (2.821%) | 1.616 (2.930%) | 1.665 (1.216%) | 2.126 (12.725%) |
| ReLD | 1.623 (5.390%) | 1.591 (1.987%) | 1.607 (2.357%) | 1.645 (*) | 1.886 (*) |

# G  ABLATION STUDY RESULTS

The ablation results for the architectural components are summarized in Table 10. The models ReLD w/o IDT & FF, ReLD-7Layer, and ReLD are trained following the settings described in Section 4.1, while all other models are trained under the same conditions as the POMO baseline. To further assess the impact of the distance heuristic on larger-scale instances, we train ReLD with and without distance heuristic using the training setup from ICAM (Zhou et al., 2024a), which primarily samples instance sizes from Uniform(100, 500). The results of this experiment are presented in Table 11.

**ReLD Improves Generalization Performance to Larger Size.** Integrating the feed-forward layer with identity mapping leverages their complementary strengths, leading to significant improvements in both in-distribution performance and out-of-distribution (OOD) generalization across problem sizes (POMON vs. POMON+IDT+FF). Additionally, incorporating the distance heuristic and training with varying attributes further enhances generalization, albeit with a minor trade-off in in-distribution performance. The combination of identity mapping and the feed-forward layer delivers consistent improvements across all problem scales, effectively narrowing the performance gap as problem size increases. These results underscore the effectiveness of the proposed architectural modifications in enhancing both scalability and adaptability.

**Where and How to Introduce Extra Parameters Matter.** We explore three additional ways to increase the number of parameters in the decoder: (1) +FF$_{qk}$: replacing the query and key transformations in the MHA module with feed-forward networks, (2) +FF$_{qkv}$: replacing the query, key, and value transformations in the MHA module with feed-forward networks, and (3) +MHA: adding an additional MHA layer on top of the original one. Our findings reveal that simply increasing the number of parameters does not necessarily improve performance. In fact, certain modifications (+FF$_{qk}$ and +FF$_{qkv}$) degrade performance significantly. This highlights the importance of thoughtfully determining where and how additional parameters are introduced into the decoder.

**Why Identity Mapping is Effective for Size Generalization?** The effectiveness likely stems from its ability to explicitly preserve context information, such as the location of the last selected node. As problem sizes grow, the decoder must aggregate a vast amount of global and local information into a fixed-size query vector, which can result in the loss of crucial locality information (Alon & Yahav, 2020). Identity mapping mitigates this issue by reinforcing the positional information of the last visited node, enabling the model to better distinguish local interactions in larger graphs.

**Adding Additional Encoder Layers to ReLD Does Not Benefit Size Generalization.** We also attempted to enhance the encoder capacity in ReLD by adding an additional layer (ReLD-7layer). Unfortunately, while this led to a slight improvement in in-distribution performance, it had no positive effect on OOD generalization, and in some cases, even worsened it. This supports the hypothesis that, with the same decoder structure, the encoder's learning task for small-scale instances has limited overlap with that for larger-scale problems, as indicated by the intrinsic complexity gap. As a result, improving the encoder alone does not necessarily enhance OOD performance due to this misalignment. These findings suggest that efforts should focus on improving the decoder structure to simplify the encoder's task and reduce the gap between the encoder's tasks across different scales.

**Distance Heuristic is Crucial.** It can be observed that the distance heuristic plays a crucial role in enabling generalization across scales. When distance information is explicitly leveraged, models consistently achieve better performance across all problem sizes. For instance, ReLD with distance

heuristic outperforms its counterpart without distance heuristic in all configurations, particularly on larger-scale instances such as CVRP5000 and CVRP7000. Notably, even without leveraging distance information, ReLD remains highly competitive with the baselines, particularly as problem sizes increase. This highlights the superiority of the proposed decoder modifications.

Table 10: Ablation study of components in ReLD.

| Method | CVRP100 Gap | CVRP200 Gap | CVRP500 Gap | CVRP1000 Gap |
|---|---|---|---|---|
| POMO | 1.004% | 3.403% | 11.135% | 110.632% |
| POMO+IDT | 0.936% | 3.096% | 9.349% | 23.508% |
| POMON | 0.947% | 3.864% | 13.008% | 33.262% |
| POMON+FF$_{qk}$ | 0.889% | 3.907% | 15.589% | 169.461% |
| POMON+FF$_{qkv}$ | 0.837% | 3.894% | 16.314% | 245.524% |
| POMON+MHA | 1.129% | 4.478% | 15.809% | 47.132% |
| POMON+IDT | 0.909% | 3.623% | 10.932% | 20.687% |
| POMON+FF | **0.663%** | 3.142% | 12.901% | 35.589% |
| POMON+IDT+FF | 0.682% | 2.640% | 8.185% | 18.076% |
| ReLD w/o IDT&FF | 1.063% | 1.960% | 4.101% | 9.623% |
| ReLD-7Layer | 0.921% | **1.638%** | 3.055% | 6.834% |
| ReLD | 0.960% | 1.690% | **3.015%** | **6.771%** |

Table 11: Impact of training on larger-scale instances. "w. dist" indicates that distance information is explicitly incorporated into the model, while "w/o dist" denotes its absence. "Single-trajec." refers to inference using a single starting point, "100-traject." involves rolling out the top 100 starting points at the first step, and "full-traject." utilizes the total number of possible trajectories, as in POMO. Augmentation is disabled for all models.

| Method | CVRP100 Obj. | CVRP1000 Obj. | CVRP2000 Obj. | CVRP5000 Obj. | CVRP7000 Obj. |
|---|---|---|---|---|---|
| HGS | 15.56 | 36.29 | 57.20 | 126.20 | 172.10 |
| GLOP-G | **15.65** | **37.09** | 63.02 | 140.35 | 191.20 |
| LEHD greedy | 16.22 | 38.91 | 61.58 | 138.17 | - |
| BQ greedy | 16.07 | 39.28 | 62.59 | 139.84 | - |
| ICAM w. dist single-trajec. | 16.19 | 38.97 | 62.38 | 140.25 | - |
| ICAM w. dist full-trajec. | 15.94 | 38.42 | 61.34 | 136.93 | - |
| ReLD w/o dist single-trajec. | 16.28 | 38.83 | 61.37 | 135.30 | 183.79 |
| ReLD w/o dist 100-trajec. | 15.98 | 38.41 | 60.78 | 134.45 | 182.75 |
| ReLD w. dist single-trajec. | 16.26 | 38.69 | 60.41 | 129.66 | 173.91 |
| ReLD w. dist 100-trajec. | 15.97 | 38.28 | **59.84** | **128.79** | **172.83** |

# H  RESULTS OF FINE-TUNING ON LARGE SCALE INSTANCES

We fine-tune the ReLD model checkpoint from the 70th epoch (before the learning rate decay) for 1 epoch (5000 batches). The instances are genenarated with size $N$ sampled from Uniform(100, 500). The expected route size (average number of nodes in a route) is sampled from triangular distribution $T(8, 16, 45)$. The learning rate is decayed by a factor of 0.1 at the 2500th batch. Batch size is set to $[120 \cdot (\frac{100}{N})^{1.6}]$, around 150,000 instances are utilized in total for fine-tuning. Other settings remain identical to training settings. For baseline, we train ELG under the same fine-tuning setting, and the results are presented in Table 12. It can be seen that ReLD also achieves better fine-tuning performance in both in-distribution data (100, 200, and 500) and out-of-distribution data (1000).

Table 12: Effects of fine-tuning on instances from Uniform(100, 500).

| Method | CVRP100 Gap | CVRP200 Gap | CVRP500 Gap | CVRP1000 Gap |
|---|---|---|---|---|
| ELG | 1.156% | 4.150% | 8.527% | 15.054% |
| ELG fine-tune | 1.895% | 1.841% | 2.281% | 5.631% |
| ReLD | 1.143% | 2.036% | 3.623% | 7.156% |
| ReLD fine-tune | 1.148% | 1.675% | 1.952% | 3.947% |

# I    ReLD Benefits from More Decoder Layer

To explore the potential gains of further increasing decoder capacity, we introduce ReLD-Large, which stacks an additional decoder block on top of the original ReLD decoder while preserving the property of static keys and values.

In specific, the original ReLD decoder can be viewed as a transformer block with a single query. We stack another single query block on top of the original one to further transform the query vector $q_c$ obtained from Eq. (9): $z_c = \text{MHA}(q_c, H_t, H_t) + q_c$, $q'_c = z_c + \text{FF}(z_c)$. The transformed $q'_c$ will be used as input to the compatibility layer. To minimize the additional memory cost, the new block shares the same cached keys and values used in Eq. (11) (i.e., share the $W^K, W^V$ parameters with the MHA module in Eq. (11)). To maintain the total number of transformer blocks consistent with the original ReLD, we reduce the number of encoder blocks in ReLD-Large to 5.

The results are presented in Table 13. Both models use 100-trajectories with instance augx8 during inference. Notably, these experiments are conducted on different hardware than specified in the paper, resulting in ReLD's runtime differing from the values reported in Table 3. It can be observed that increasing the decoder's complexity further enhances model performance across all sizes, albeit with greater computational overhead.

Table 13: Results of stacking an additional layer in the decoder.

| Method | CVRP100 Gap (time) | CVRP200 Gap (time) | CVRP500 Gap (time) | CVRP1000 Gap (time) |
|---|---|---|---|---|
| ReLD | 0.959% (2.16m) | 1.654% (0.11m) | 2.975% (0.48m) | 6.757% (1.95m) |
| ReLD-Large | 0.891% (3.60m) | 1.527% (0.16m) | 2.678% (0.73m) | 6.265% (2.89m) |

