# OpenReview forum: "Rethinking Light Decoder-based Solvers for Vehicle Routing Problems"
_ICLR.cc/2025/Conference — ICLR 2025 Poster_

### Official Review · Reviewer_EM2N · 2024-10-24

**Soundness:** 3
**Presentation:** 3
**Contribution:** 3
**Rating:** 6
**Confidence:** 4

**Summary:**

This paper discusses the limitations of the Heavy Encoder-Light Decoder based model, which is commonly used in combinatorial optimization. Specifically, it addresses the problem where the encoder must embed all possible contextual information required during the decoding process into a single embedding, resulting in excessively high information density. It also highlights how the decoder fails to utilize this information. As a solution, the authors propose ReLD. The proposed method is validated through experiments on CVRP and VRP variants, demonstrating its effectiveness. In particular, the method showed superior performance compared to existing Light Decoder-based solvers when faced with out-of-distribution (OOD) problems.

**Strengths:**

- This paper analyzes the limitations of the Light Decoder-based model and demonstrates the complexity of the encoded information and lack of generalization performance through experiments.

- This paper proposes a novel approach by modifying the decoder to overcome the issues inherent in Light Decoder-based models, and demonstrates the superiority of the proposed method through experiments.

- In the CVRP generalization experiments, it showed promising results than existing Light Decoder-based models and performed well across various VRPs.

**Weaknesses:**

- There are unclear parts in the equations and content of Section 3 Methodology. It also does not match well with Figure 1(e). It is unclear whether $Q$ in Figure 1 (e) is the same as $q_c$ in Equation (11) or not. And It is difficult to understand the structure of the proposed neural network in Equation (7) to (11). For example, it is not clear how $q_c$ in Equation (11) is used. Relevant questions can be found in the Question section.

- In the experiments, it is unclear whether the improvement in VRPs solution accuracy is due to the addition of the feedforward network itself or simply the increase in decoder parameters caused by adding FF. It seems necessary to conduct a comparative evaluation with a decoder where the parameters are increased equivalently without FF. Furthermore, a more solid logical explanation is needed on how the FF helps overcome the limitations of the light decoder.

**Questions:**

- How $q_c$ in Equation (11) is used?

- What are $W_S^OW_S^V$ in Equation (8)? Does this imply performing value projection followed directly by output projection? As far as I know, such an operation does not exist in the original transformer. Could you provide a more detailed explanation regarding this part?

- The modification presented in this paper adds an FF layer to the POMO decoder, and the decoders in models like POMO operate almost as many times as the number of nodes in an auto-regressive manner. Therefore, it is expected that the model modification proposed in this paper will increase both training and inference (optimization) times. However, in line 361 of the paper, it states that the additional step-wise running time is independent of the number of nodes and is computationally efficient. Could you please provide a more detailed explanation on this? If possible, it would be helpful to provide numbers on the changes in training and inference times due to the model modification.

- How effective is the Distance Heuristic($-log(dist_i)$) in Equation 12? Is there any information on how the model performs if this heuristic is removed?

- In Table 3, the CVRP100 gap for POMO augx8 is 1.004%, which shows a significant difference from the 0.32% gap for CVRP100 presented in the original POMO paper. Could you explain the reason for this discrepancy?

- In Table 3, the addition of the ff layer appears to provide little benefit for CVRP100. Why does adding the ff layer to the decoder not contribute to performance improvement for a problem size 100?

- As a minor comment, the font size in Tables 3 and 4 is too small.

---

### Official Review · Reviewer_nQhH · 2024-10-27

**Soundness:** 3
**Presentation:** 2
**Contribution:** 2
**Rating:** 6
**Confidence:** 4

**Summary:**

The authors first address the challenges of light decoder-based approaches for vehicle routing problems (VRPs). Because the traditional approach relies on static embeddings that must capture complex information within a single set of representations, it is difficult for the simplistic decoder to fully leverage this information, particularly in out-of-distribution (OOD) scenarios (such as generalizing to larger instances or different VRP variants). Enhancements to the decoder is thus introduced, such as adding an identity mapping and a feed-forward layer, to mitigate this issue. Experimental results demonstrate that this adjustment improves both in-distribution and OOD generalization performance, narrowing the gap between light and heavy decoder paradigms.

**Strengths:**

The authors effectively highlight a key issue with static embeddings: the gap between the information needed for optimal decoder performance and the information stored in the context vector. This insight is both logical and significant, emphasizing a critical area for future improvements in this field. The research in this direction holds promising implications for advancing VRP solutions.

**Weaknesses:**

(1) The issue of the overly complex context vector is not directly addressed. The authors’ solution is somewhat simplistic, adding only extra structures to the decoder without any modification to the encoder or main inference process. This solution still has the same 'complex context vector' problem as described in "Gap between Policy Formulations" subsection (LINE 195). Given this limited change, the improvement in performance, such as in CVRP100, is also marginal.

(2) One of the paper's main claims is that their method improves out-of-distribution (OOD) performance. However, the practicality of the approach based on zero-shot generalization is unclear. Why is it necessary for a model trained on N=100 cases to perform well on N=1000 cases? There are many established methods to address OOD problems, such as fine-tuning before inference, tree search or active learning during inference. The proposed modifications should ideally be evaluated in these more practical and realistic settings rather than in zero-shot scenarios.

(3) With additional parameters in the decoder, the training burden likely increases. The training details are largely missing, particularly regarding how the proposed modifications affect training time and resources. The authors should also compare their approach to other potential modifications that increase parameters in the decoder, such as adding an additional decoding layer, in terms of training efficiency and resource requirements, as well as the solver performance.

**Questions:**

These are not questions, but rather minor shortcomings related to the paper’s writing.

(1) There are too many versions of ReLD introduced in the paper, and it is difficult to follow what each version represents.

(2) Shouldn't Figure 1 also reflect the changes described in Section 3.2, Powerful Query? Currently, it seems to only illustrate the changes discussed in Section 3.1, Direct Influence of Context.

(3) Table 5 contains several mis-copied numbers and incorrect placement of bold formatting.

---

### Official Review · Reviewer_p2Jk · 2024-11-04

**Soundness:** 3
**Presentation:** 3
**Contribution:** 3
**Rating:** 6
**Confidence:** 3

**Summary:**

This paper studies the limitations of light decoder-based solvers for Vehicle Routing Problems (VRP). Modern RL methods for VRP typically employ light decoders for solution generation due to their efficiency. However, the authors think the light decoders may not capture the problem structure well. So they proposed Revised Light Decoder (ReLD) which modified the original light decoder and make it contain richer information.

The experiment results show that their framework can improve the current state-of-the-art methods.

**Strengths:**

1. This paper approaches the deep learning for VRP problem from a new perspective: make the decoder contain richer information.
2. The modification is not complicated and the results looks good.

**Weaknesses:**

1. The scalability issue still exists. There are no large-scale experiments conducted.

**Questions:**

1. Do you have any results on the large-scale instances, e.g., CVRP-10000?
2. Can this method be transferred to TSP or other CO problems? Do you have any preliminary results?

---

### Official Review · Reviewer_NKcN · 2024-11-05

**Soundness:** 2
**Presentation:** 3
**Contribution:** 3
**Rating:** 6
**Confidence:** 4

**Summary:**

The paper revisits light decoder-based solvers for VRPs, recognized for their efficiency but limited in generalization to larger or varied problem instances. The authors attribute this limitation to the handling of static embeddings, which creates high information density in the encoder, overwhelming the simplistic decoder. To overcome these challenges, they propose an enhanced decoder structure, incorporating identity mapping and feed-forward layers to effectively boost the decoder’s capacity and improve generalization performance. The authors perform experiments to demonstrate that ReLD achieves better generalization performance on both in-distribution and out-of-distribution tasks, closing the performance gap with heavier decoders while maintaining computational efficiency.

**Strengths:**

1. The paper is well-written and easy to follow
2. A thorough empirical analysis is conducted to validate the potential limitation of current light decoder-based solvers
3. A simple but effective modification is performed to improve the decoder part of the current decoder-based solvers
4. Experiments are conducted on many datasets covering different distributions, problem sizes, and problem classes.

**Weaknesses:**

The analysis in this paper is largely 'end-to-end,' with a strong reliance on empirical results presented towards the conclusion. Several concerns regarding the encoder and decoder architectures are raised, and it may be beneficial to adopt a more direct investigative approach into these components.

Additionally, the limitations of the current light decoder-based model are inferred from empirical experiments conducted solely on CVRP problems, focusing on LEHD and POMO models. A broader analysis including more models with both light and heavy decoder architectures would provide a more comprehensive foundation for the conclusions drawn.

The insights presented offer valuable guidance on improving the decoder architecture to address limitations associated with overly simplified decoders. This work implements a minor modification in this direction; however, the extent to which further increases in model complexity would yield additional performance gains remains uncertain. This issue points to a trade-off between model performance and efficiency, though an optimal balance between the two has yet to be determined.

Overall, I appreciate the discussions and architectural considerations raised by this paper concerning VRP model design. Currently, I lean toward borderline acceptance.

**Questions:**

Besides the concerns raised in the weakness part, I have the following additional questions:

1. In Table 5, please check whether all the numbers you are reporting are correctly documented. The bolded number in OVRPLTW and the bolded number in OVRPBLTW are either too big or too small and do not correspond to the reported gaps.
2. the figure 1 could be further improved. E.g. the font size and the caption.

---

### Official Review · Reviewer_SfnF · 2024-11-06

**Soundness:** 3
**Presentation:** 2
**Contribution:** 2
**Rating:** 6
**Confidence:** 4

**Summary:**

This paper presents an analysis of light decoder-based solvers for VRP, specifically addressing the challenges of generalization to out-of-distribution (OOD) problem instances. By identifying limitations due to the reliance on static embeddings, the authors propose a modified approach, ReLD (Rethinking Light Decoder), which incorporates identity mapping and a feed-forward layer to enhance the decoder’s capacity. The proposed model demonstrates improved OOD performance across a variety of VRP instances, narrowing the performance gap between light and heavy decoder approaches.

**Strengths:**

The paper provides a detailed breakdown of the light decoder’s limitations in VRP, particularly the static embeddings’ burden on the encoder.

ReLD addresses an important need in VRP research generalization across problem scales.

The modifications retain the light decoder’s computational efficiency, which could be advantageous for applications needing faster routing solutions without the computational load of heavy decoders.

**Weaknesses:**

Limited Scalability for Large Instances: ReLD struggles to compete with heavy decoder architectures on very large instances, such as CVRP1000. This limitation suggests that ReLD’s current modifications might not be sufficient for all scales of VRP.

The proposed modifications are relatively minor adjustments. While effective, they lack substantial novelty within the machine learning field.

While this paper makes a valuable contribution by revisiting the light decoder paradigm and identifying limitations in current architectures, its primary innovations are modest. The architectural modifications, though effective, are straightforward and may not sufficiently address scalability issues, particularly in very large instances or complex real-world VRP variants. Moreover, the method is not showing SOTA results on the biggest problem instances, which is assumed to be the main advantage of the method.

**Questions:**

How does ReLD manage complex or dynamic VRP constraints, such as real-time updates or varying demands?
How could larger decoder modifications enhance OOD performance?

---

### Comment · Area_Chair_DccF · 2024-11-25
**Important: Please Review Rebuttals and Update Reviews as Needed**

Dear Reviewers,

Thank you for your hard work and dedication to providing thoughtful reviews for this year’s ICLR submissions. Your efforts play a vital role in maintaining the conference’s high standards and fostering meaningful discussions in the community.

As we are close to the end of the discussion phase, I kindly urge you to read the authors’ responses and reevaluate your reviews carefully, especially if they address your concerns. If you find that the authors have resolved the issues or clarified misunderstandings, please consider adjusting your comments and scores accordingly. This ensures fairness and gives each submission the opportunity to be judged on its merits.

Your continued commitment is greatly appreciated—thank you for contributing to the success of ICLR!

---

### Meta-Review · Area_Chair_DccF · 2024-12-19

**Metareview:**

This paper analyzes the limitations of light decoder-based methods for solving Vehicle Routing Problems (VRPs). The authors clearly identify that static embeddings in the encoder result in dense information that the light decoder struggles to utilize, especially when generalizing to larger or more complex VRP instances. Their proposed solution, ReLD, introduces simple yet effective modifications to the decoder, specifically identity mapping and a feed-forward layer. These enhancements alleviate the burden on the encoder and improve the decoder’s capacity to handle context-specific information. This approach maintains the computational efficiency of light decoder-based methods while improving out-of-distribution (OOD) performance.

The strengths of the paper lie in its clear problem diagnosis, logical modifications, and thorough empirical evaluation. The authors demonstrate that ReLD significantly improves generalization performance on large-scale and complex VRP variants, narrowing the gap between light and heavy decoder paradigms. Notably, ReLD performs well on real-world datasets and scales effectively to instances as large as CVRP16K. The paper’s insights provide a valuable contribution to neural combinatorial optimization, offering a balance between performance and efficiency.

**Additional Comments On Reviewer Discussion:**

The reviewer discussion was constructive and led to meaningful improvements to the paper. Reviewers raised concerns about scalability, clarity in the methodology, and the novelty of the modifications. The authors addressed these points by providing additional experiments on large-scale instances, clarifying key equations, and demonstrating the effectiveness of their decoder enhancements compared to other parameter-increasing strategies.

---

### Decision · Program_Chairs · 2025-01-22

Accept (Poster)